# Computed Tomography Morphology of Affected versus Unaffected Sides in Patients with Unilateral Primary Acquired Nasolacrimal Duct Obstruction

**DOI:** 10.3390/jcm12010340

**Published:** 2023-01-01

**Authors:** Pei-Yuan Su, Jia-Kang Wang, Shu-Wen Chang

**Affiliations:** 1Ophthalmology Department, Far-Eastern Memorial Hospital, New Taipei City 220, Taiwan; 2School of Medicine, Fu-Jen Catholic, New Taipei City 242062, Taiwan

**Keywords:** primary acquired nasolacrimal duct obstruction, bony nasolacrimal duct, computed-tomography morphology

## Abstract

Background: This study aimed to describe the anatomical details of the bony nasolacrimal duct (BNLD) and adjacent nasal structures by analyzing computed tomography (CT) images, and to investigate their effects on the development of primary acquired nasolacrimal duct obstruction (PANDO). Methods: A total of 50 patients with unilateral PANDO who underwent dacryocystorhinostomy, with a mean age of 57.96 years, were included. The preoperative CT images were reviewed to measure the anteroposterior and transverse diameters of the BNLD at the entrance and exit levels, as well as the minimum transverse diameter along the tract. The sagittal CT images were analyzed to classify the shape of the bony canals into columnar, funnel, flare, and hourglass. The associated paranasal abnormalities, including nasal septum deviation (NSD), sinusitis, angle between the bony inferior turbinate and medial wall of the maxillary sinus, and mucosal thickness of the inferior turbinate, were investigated. Results: Fifty CT images were analyzed, and all parameters measured on both sides of the BNLD were not significantly different between the PANDO and non-PANDO sides, except for the minimum transverse diameter, which was significantly smaller on the PANDO side (*p* = 0.002). Columnar-shaped BNLD was the most common on both sides. No significant difference was observed in the incidence of paranasal abnormalities between sides; however, deviation of the septum toward the non-PANDO side was more common (67.9%). Conclusions: A small minimum transverse diameter of the BNLD may be a risk factor for PANDO. The association between nasal abnormalities and PANDO was not remarkable.

## 1. Introduction

Epiphora due to nasolacrimal duct obstruction (NLDO) is a common ophthalmic problem that accounts for approximately 3% of clinical visits [1]. The obstruction of the lacrimal drainage system can be congenital or acquired. Congenital NLDO, which presents symptoms soon after birth, commonly results from a persistent membrane at the valve of Hasner. Acquired NLDO, which manifests later in life, can be classified into primary or secondary NLDO. Common causes of secondary acquired NLDO include facial trauma or surgery, neoplasm, sarcoidosis, and Wegener’s granulomatosis [2].

Primary acquired NLDO (PANDO) is frequent among older women and is usually bilateral. PANDO is characterized by gradual chronic inflammation and fibrosis along the nasolacrimal duct, leading to obstruction of the drainage system [1,3]. Although PANDO is idiopathic, many predisposing factors, such as conjunctival infection, nasal disease, hormone fluctuation, sinusitis, female sex, smoking, and topical glaucoma medication, have been suggested [1,4,5,6].

Anatomical variation of the bony nasolacrimal duct (BNLD) is also a crucial risk factor for PANDO. Narrowing of the bony canal may cause stasis of the tear flow, accumulation of debris and inflammatory products, adhesion and fibrosis of the internal nasolacrimal duct mucosa, and finally obstruction of the drainage pathway. Some studies have compared the diameter of the BNLD among normal individuals of different age groups, sexes, and ethnicities [7,8,9]. One study reported a smaller diameter of the BNLD in the PANDO group than in the control group; however, other studies have reported no significant difference in the BNLD diameter between the groups [7,10,11,12,13]. Therefore, the results on the role of BNLD morphology in the development of PANDO are controversial. Because of the anatomical proximity, the lacrimal drainage pathway may be affected by pathology in the nasal cavity and paranasal sinuses [14]. Relationships between PANDO and paranasal sinusitis, nasal septum deviation (NSD), the angle between the bony inferior turbinate and nasal lateral wall at the end of the nasolacrimal duct, and structural abnormalities of the sinonasal cavity have been studied [15,16,17,18]; however, the results are inconclusive.

To determine the risk factors for PANDO, we compared the morphology of the BNLD and sinonasal abnormality of the affected (PANDO) and unaffected (non-PANDO) sides of our patients with unilateral PANDO by reviewing computed tomography (CT) images. Observations included the bony canal diameter at different levels, aeration inside the canal, the canal shape in sagittal view, NSD severity, the presence of sinusitis, and inferior turbinate structure at the distal end of the BNLD.

## 2. Materials and Methods

### 2.1. Participants and Ethics

This study was approved by the Institutional Review Board of Far Eastern Memorial Hospital, New Taipei City, Taiwan (NO. 111038-E), and the tenets of the Declaration of Helsinki were followed. The preoperative orbital CT scans of all patients with unilateral PANDO who presented to our ophthalmology clinic from December 2019 to May 2021 were retrospectively reviewed. The diagnosis of PANDO was made through lacrimal irrigation and probing under topical anesthesia in the outpatient department. Once the patients agreed to receive further surgical intervention, orbital CT scans were obtained to rule out lacrimal sac tumors or nasal pathologies that could cause secondary NLDO. Patients with congenital NLDO; bilateral PANDO; NLDO secondary to trauma, tumor, or orbital irradiation; or a history of sinusitis or lacrimal surgery were excluded. A total of 50 patients with unilateral PANDO were included, and CT scan images of the PANDO and non-PANDO sides were evaluated separately.

### 2.2. Data Retrieval and Processing

CT scans were performed using a 64-slice high-speed scanner (SOMATOM Definition AS; Siemens, Munich, Germany). Contiguous 2 mm axial and sagittal images were obtained parallel and perpendicular to the orbital floor. Anatomical measurements were made by a single investigator by using the caliber tools of the viewer through bone windows.

To evaluate BNLD morphology, the anteroposterior and transverse diameters of the inner bony canal in axial view were measured at three levels: the entrance point at the inferior orbital margin, the exit point at the opening of the inferior meatus, and the minimum transverse diameter along the bony canal (Figure 1). In the sagittal view, the longest section of the BNLD, in which proximal and distal ends were visible, was selected for evaluating the shape of the bony canal. We classified the BNLD shape into four types according to the location of the narrow point (the minimum anteroposterior diameter) within the bony canal (Figure 2). In the funnel type, the narrow point was located near the exit level of the BNLD, whereas in the flare type, the narrow point was located at the BNLD entrance. In the hourglass type, the narrow point was located between the exit and entrance levels. In the columnar type, the anteroposterior diameters of the inner BNLD were evenly distributed along the pathway.

To evaluate the anatomy of the part of the nasal cavity into which the nasolacrimal duct drained, a single coronal section of the CT image corresponding to the most distal part of the BNLD was selected. Measurements included the NSD angle, the angle between the bony inferior turbinate and medial wall of the maxillary sinus, and transverse mucosal thickness of the inferior turbinate (Figure 3). The NSD angle was measured by drawing a vertical line from the crista galli to the nasal crest of the maxillary bone and another line to the maximum deviation of the nasal septum.

### 2.3. Statistical Analyses

Statistical analyses were performed using SPSS 23.0. The independent sample *t* test, chi-squared test, and Pearson’s correlation were used. A *p* value of <0.05 was considered statistically significant.

## 3. Results

### Patient Characteristics and Measurement Results

A total of 50 patients with unilateral PANDO (4 men and 46 women) and a mean age of 57.96 years (ranging from 35 to 77 years) were included. PANDO laterality was equally distributed among the patients (25 left and 25 right sides).

The BNLD measurement results of the PANDO and non-PANDO sides are listed in Table 1. The mean anteroposterior diameter and transverse diameter of both sides were not significantly different over the entrance level (*p* = 0.439 and 0.188, respectively). No significant differences in the mean anteroposterior diameter and transverse diameter were observed between the sides at the exit level (*p* = 0.357 and 0.085, respectively). However, significantly smaller minimum transverse diameters were observed on the PANDO sides (mean ± SD = 3.63 ± 1.08 mm) than on the non-PANDO sides (mean ± SD = 4.07 ± 1.13 mm; *p* = 0.002), and same results were obtained for both sexes (*p* = 0.016 for female and 0.023 for male). We also found that mean minimum transverse diameters were significantly larger in males on both sides (*p* = 0.014 on PANDO sides and 0.036 on non-PANDO sides. Comparisons of the BNLD measurements between sexes are summarized in Table 2. All measured parameters were not correlated with age for either the PANDO or non-PANDO sides; the results are summarized in Table 3.

The analysis of the BNDL shape using sagittal CT images of the orbits revealed that the most common shape of the BNLD was columnar (accounting for 48% and 66% of the PANDO and non-PANDO sides, respectively), followed by flare, funnel, and hourglass. No significant difference in the BNDL shape was observed between the PANDO and non-PANDO sides (*p* = 0.205); the results are listed in Table 4.

The CT images revealed that 28 patients (56%) had NSD, with a mean angle of deviation of 5.38° ± 6.38° (ranging from 3.2° to 26.3°). Of them, 19 patients (67.9%) had their septum deviated to the non-PANDO side. The measurements of the coronal section corresponding to the exit level of the BNLD revealed that the mean angle between the bony inferior turbinate and medial wall of the maxillary sinus was 53.84° ± 17.94° on the PANDO sides and 54.73° ± 15.38° on the non-PANDO sides. No significant difference in the mean angle was observed between the sides (*p* = 0.295). The mean highest transverse mucosal thickness of the inferior turbinate from the same coronal section was 8.47 ± 1.60 mm on the PANDO sides and 8.25 ± 1.97 mm on the non-PANDO sides. No significant difference in the mean highest transverse mucosal thickness was observed between the sides (*p* = 0.196). The CT images revealed that 10 patients (20%) had an opacity of the paranasal sinus on the PANDO sides, whereas 7 patients (14%) had an opacity of the paranasal sinus on the non-PANDO sides. No significant difference in opacity was observed between the sides (*p* = 0.323).

## 4. Discussion

Several studies have measured the BNLD diameter in the normal population. By measuring the epoxy resin casts of macerated skulls, Steinkogler [3] reported a transverse BNLD diameter of 4.8 mm. Janssen [7] included 100 controls in a study in the Netherlands and determined the mean minimum transverse diameter to be 3.7 mm in men and 3.35 mm in women by reviewing axial CT images. Lee [8] reported a mean transverse diameter of 4.5 mm and a minimum transverse diameter of 3.2 mm among 228 Korean patients without NLDO. Takahashi [12] reviewed the CT images of 100 sides of 50 Japanese patients without NLDO and reported a minimum transverse diameter of 4.8 mm. Fasina [11] measured the minimum BNLD diameter of 401 Nigerian adults using CT images and reported a diameter of 3.52 mm in men and 3.36 mm in women. In our study of Taiwanese adults, the minimum transverse diameter was 4.07 mm on the non-PANDO sides, which was higher than those in the aforementioned study groups except for the Takahashi group’s results for the Japanese population. Although the prevalence of PANDO is lower among Africans than among Caucasians and Asians, the impact of ethnicity on PANDO is not conclusive from the perspective of BNLD size.

A small diameter of the bony canal is one of the proposed etiological factors contributing to NLDO. One study calculated the flow resistance in a tube by using an equation and reported that a 0.3 mm reduction in tube diameter increases resistance to the water flow by 1.38 times [9]. In our study, the anteroposterior and transverse diameters of the BNLD were not significantly different between the PANDO and non-PANDO sides at either the entrance or exit levels. However, the minimum transverse diameter on the PANDO sides (3.63 ± 1.08 mm) was significantly smaller than that on the non-PANDO sides (4.07 ± 1.13 mm; *p* = 0.002), and the difference was more than 0.3 mm. Similar results were reported by Janssen et al. [7], wherein the minimum transverse diameter of the BNLD was smaller in patients with PANDO than in controls (3.0 vs. 3.5 mm, *p* = 0.001). However, the sample size of the study was small (the patient group *n* = 24, the control group *n* = 100). Another comparative study by Takahashi [12] with a larger sample size (101 patients with unilateral PANDO and 50 controls) reported that the minimum transverse diameter on the NLDO sides was 5.09 mm, that on non-NLDO sides was 4.96 mm, and that on control sides was 4.80 mm. However, no significant difference in the minimum transverse diameter was noted between the groups. Bulbul [13] enrolled 39 patients with unilateral PANDO and 36 controls and discovered that the minimum and distal transverse diameters of the BNLD in patients with PANDO were significantly smaller than those in controls (*p* = 0.04 and <0.001, respectively). However, no differences in any BNLD measurements were observed between the NLDO and non-NLDO sides in patients with PANDO. The results of previous studies varied. Though the possibility of future development of nasolacrimal duct obstruction in the presently unaffected eyes cannot be ruled out, our results suggest that eyes with smaller BNLD diameter are prone to early presence of PANDO.

Changes in the lumen along the lacrimal passage may also influence the resistance of the tear flow [8]. Takahashi [12] reported that funnel-shaped BNLD with a minimum diameter at the canal entrance was more common in patients with NLDO than in controls. Moreover, because funnel-shaped BNLD was more common among female patients with NLDO, they concluded that this shape may increase the incidence of NLDO among women. To evaluate the effect of different narrow-point locations on the formation of an obstruction, we described the shapes of the BNLD and classified them into four types, columnar, flare, funnel, and hourglass. Our study demonstrated that columnar-shaped BNLD was the most common on both PANDO and non-PANDO sides. Though the results showed that there was no statistically significant difference in proportion for all types between the two sides, we did find that noncolumnar types were more frequently observed on PANDO sides compared to non-PANDO sides (52% vs. 34%). Few studies have emphasized the relationship of the BNLD shape or location of the narrow point with NLDO occurrence. Our study provides a more detailed description of the canal shape according to location of narrowing, and supports the idea that changes in canal size along the tract despite different levels might play a role in the etiology of PANDO. Further studies should be conducted to compare these results with those of the normal population.

The relationship between the occurrence of PANDO and paranasal sinus pathology remains controversial. Kallman [19] evaluated the CT images of 23 patients with PANDO and 100 controls and reported a significantly higher rate of abnormal paranasal sinus findings, including NSD and ethmoidal opacification, in the PANDO group (87% vs. 63%). Dikici [18] investigated the CT images of 37 patients with PANDO and 37 controls and discovered a significant relationship between PANDO and axial location or NSD angle. Habesoglu [17] studied 41 patients with unilateral PANDO and reported no significant difference in the rates of NSD and ethmoidal sinusitis between the PANDO and control groups. Yazici [15] compared 40 unilateral PANDO patients with 71 controls and discovered no significant difference in the incidence of paranasal sinus abnormalities, including NSD location, NSD angle, and sinusitis, between the PANDO sides and the non-PANDO sides or controls. A correlation was observed only between the NSD and PANDO sides. Our study revealed that 56% of the patients with unilateral PANDO had NSD. Moreover, the nasal septum deviated more toward the non-PANDO sides (67.9%), a result contradictory to that of Yazici’s study. However, the incidence of paranasal sinusitis in our study was not significantly different between the PANDO and non-PANDO sides (*p* = 0.323). The varying results may be due to the lack of a standard definition for all paranasal sinus pathologies, such as NSD location or sinusitis severity. Previous studies have demonstrated that NSD is one of the common causes of dacryocystorhinostomy failure [20,21], which may be attributed to the obstruction of surgical access or postoperative adhesion leading to ostium closure. The role of NSD in the development of PANDO remains inconclusive.

Narrowing of the nasal structure at the distal opening of the nasolacrimal duct is considered a predisposing factor for the occurrence of PANDO. Yazici [15] discovered no difference in mucosal thickness and the lateralization angle of the inferior turbinate between the NLDO and non-NLDO groups. Gul [16] reported that the mean angle between the bony inferior turbinate and medial wall of the maxillary sinus was significantly narrower on the affected sides than on the unaffected sides in the PANDO group (56.2° vs. 58.6°, *p* = 0.01). Dikici [18] compared the angle width between the bony inferior turbinate and medial wall of the maxillary sinus between patients with PANDO and controls and discovered that the angle was significantly wider on both sides in patients with PANDO (*p* = 0.007 for the right sides and *p* = 0.006 for the left sides). In our study, no significant difference in either the mucosal thickness of the inferior turbinate or angle between the bony inferior turbinate and medial wall of the maxillary sinus was observed between the PANDO and non-PANDO sides (*p* = 0.295 and 0.196, respectively). The nasolacrimal duct opens into the vault of the meatus located underneath the inferior turbinate, but it may extend further down and open at various positions in nasal lateral wall. Therefore, we speculated that the location selected for the measurement of the inferior turbinate angle width may not truly reflect the impact of the narrow nasal structure on the drainage of tear duct.

Our study was limited by the retrospective study design and the lack of a normal control group for comparison. Moreover, the possibility of PANDO development on unaffected sides in the future cannot be ruled out. Future studies using three-dimensional CT image reconstruction may be needed to demonstrate the details of the bony structure of nasolacrimal duct more precisely.

## 5. Conclusions

In conclusion, columnar-shaped BNLD, with no obvious narrowing along the path on either side, was more frequent among our patients with unilateral PANDO. A smaller minimum transverse diameter of the bony canal was observed on the PANDO sides than on the non-PANDO sides, which might indicate a risk factor for NLDO. The nasal septum deviated more toward the non-PANDO sides. No significant difference in the lateralization of the bony inferior turbinate, mucosal thickness of the inferior turbinate, or incidence of sinusitis was observed between the sides. The association between nasal abnormalities and NLDO remains unclear. Our study provides information on anatomical details of the BNLD and associated nasal structures, thereby improving the understanding of PANDO pathophysiology.

## Figures and Tables

**Figure 1 jcm-12-00340-f001:**
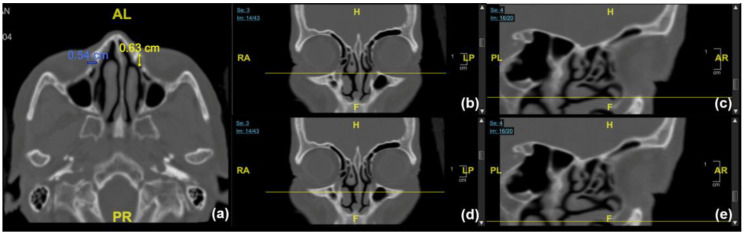
(**a**) Anteroposterior and transverse diameters of the inner bony canal in axial view. The diameters were measured at three levels: the entrance point at inferior orbital margin (**b**,**c**), the exit point at the opening of inferior meatus (**d**,**e**), and the minimum transverse diameter along the bony canal.

**Figure 2 jcm-12-00340-f002:**
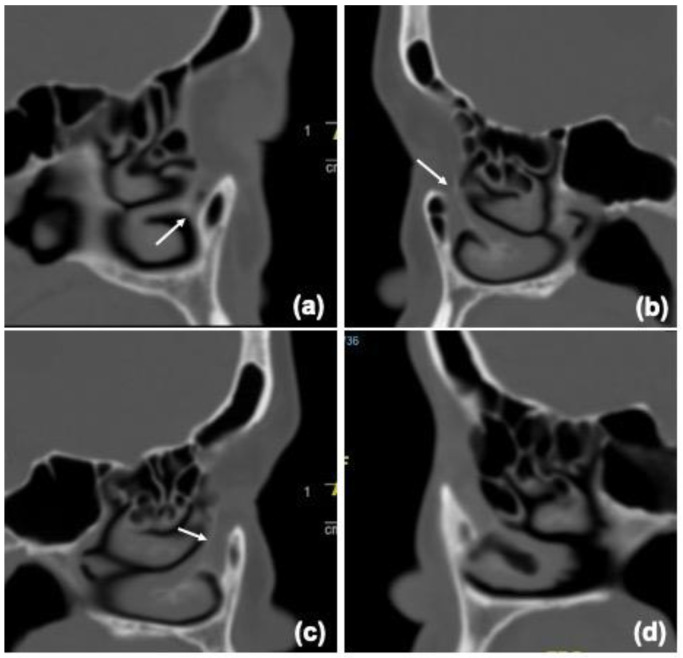
Four shapes of the BNLD. (**a**) Funnel type: narrow point near the exit level. (**b**) Flare type: narrow point near the entrance level. (**c**) Hourglass type: narrow point between the exit and entrance levels. (**d**) Columnar type: evenly distributed along the path.

**Figure 3 jcm-12-00340-f003:**
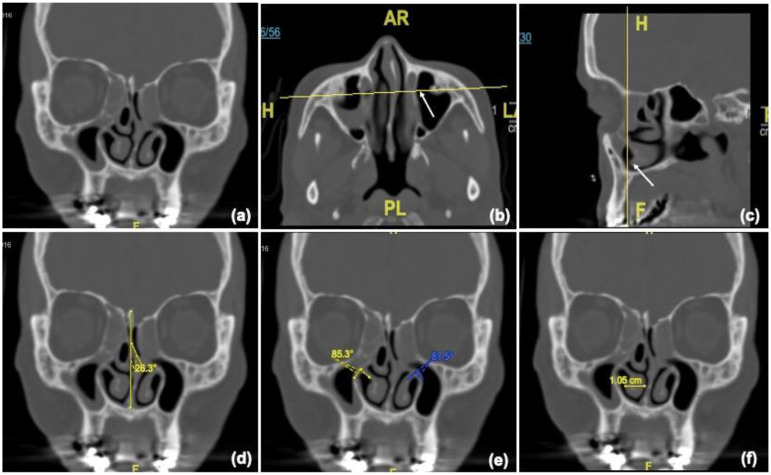
(**a**) Coronal sections corresponding to the most distal part of the BNLD were selected (**b**) from axial and (**c**) sagittal CT images. Measurements included NSD angle (**d**), angle between the bony inferior turbinate and medial wall of the maxillary sinus (**e**), and transverse mucosal thickness of the inferior turbinate (**f**).

**Table 1 jcm-12-00340-t001:** Measurements of the bony nasolacrimal duct.

/	PANDO Side		Non-PANDO Side		*p*-Value	
	A-P Diameter	Transverse Diameter	A-P Diameter	Transverse Diameter		
	Mean ± SD (mm)					
Entrance	5.77 ± 1.39	4.49 ± 1.32	5.76 ± 1.30	4.58 ± 1.31	0.439	0.188
Exit	7.06 ± 1.94	4.22 ± 1.15	7.00 ± 1.79	4.37 ± 1.05	0.357	0.085
Minimum		3.63 ± 1.08		4.07 ± 1.13		0.002 *

PANDO: primary acquired nasolacrimal duct obstruction; A-P: Anteroposterior; * Statistically significant according to independent *t* test.

**Table 2 jcm-12-00340-t002:** Comparison of the bony nasolacrimal duct between sexes.

	PANDO Side		Non-PANDO Side		*p*-Value	
Diameter	A-P	Transverse	A-P	Transverse		
	Mean ± SD (mm)		Mean ± SD (mm)			
Female (*n* = 46)						
Entrance	5.77 ± 1.39	4.37 ± 1.24	5.73± 1.29	4.46 ± 1.24	0.444	0.353
Exit	7.03± 2.01	4.19 ± 1.19	6.90 ± 1.78	4.27 ± 1.01	0.263	0.076
Minimum		3.57 ± 1.08		3.98 ± 1.11		0.016 *
Male (*n* = 4)						
Entrance	5.83 ± 1.62	5.95 ± 1.51	6.10± 1.51	5.93 ± 1.44	0.406	0.491
Exit	7.38± 0.85	4.60 ± 0.39	8.08 ± 1.65	5.48 ± 0.92	0.240	0.066
Minimum		4.70 ± 0.55		5.10 ± 0.85		0.023 *
*p*-value						
Entrance	0.474	0.062	0.331	0.066		
Exit	0.263	0.076	0.127	0.073		
Minimum		0.014 **		0.036 **		

PANDO: primary acquired nasolacrimal duct obstruction; A-P: Anteroposterior; * Statistically significant according to independent *t* test; ** Statistically significant according Mann–Whitney U test.

**Table 3 jcm-12-00340-t003:** Correlation between age and the bony nasolacrimal duct.

	Entrance Level		Minimum Transverse Diameter	Exit Level	
	A-P Diameter	Transverse Diameter		A-P Diameter	Transverse Diameter
	r (correlation coefficient)				
PANDO side	0.170	0.202	0.179	0.194	0.057
non-PANDO side	0.273	0.225	0.205	0.211	0.206

PANDO: primary acquired nasolacrimal duct obstruction; A-P: Anteroposterior; r: Pearson’s correlation coefficient.

**Table 4 jcm-12-00340-t004:** Shape of the bony nasolacrimal duct.

	PANDO Side		Non-PANDO Side	
	*n*	(%)	*n*	(%)
Columnar	24	48	33	66
Flare	17	34	11	22
Funnel	4	8	2	4
Hourglass	5	10	4	8

PANDO: primary acquired nasolacrimal duct obstruction.

## Data Availability

Not applied.

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
