# Peer review of "Computed Tomography Morphology of Affected versus Unaffected Sides in Patients with Unilateral Primary Acquired Nasolacrimal Duct Obstruction"

_jcm, 2023, doi:10.3390/jcm12010340_

Round 1

Reviewer 1 Report (New Reviewer)

1. Too lengthy citations can be rephrased. Eg: [7-13].

2. How does the authors distinguish their work from others in the literature?

3. What are the research gaps in the related studies? How this work address those issues?

4. Novelty to be stated clearly.

5. Proof reading strongly recommended.

Author Response

please see the attached file 

Reviewer 2 Report (New Reviewer)

This is a nice, very well-written paper by Sue et al about computed-tomography morphology of affected Versus unaffected sides in patients with unilateral primary acquired nasolacrimal duct obstruction. The methodology is resaonable, the results are clear and nicely described and previous publsihed papers from the literature are provided in the discussion.

Author Response

please see the attached file 

Reviewer 3 Report (New Reviewer)

In this manuscript, the authors described the anatomical details of the bony nasolacrimal duct (BNLD) and adjacent nasal structures by analyzing computed tomography (CT) images and to investigate their effect on the development of primary acquired nasolacrimal duct obstruction (PANDO). The anteroposterior and transverse diameters of the BNLD at the entrance and exit levels, and the minimum transverse diameter along the tract were measure. The shape of the bony canals were analyzed to classify into columnar, funnel, flare, and hourglass. The associated paranasal abnormalities, including nasal septum deviation (NSD), sinusitis, the angle between the bony inferior turbinate and medial wall of the maxillary sinus, and mucosa thickness of the inferior turbinate, were investigated.

However, all parameters measured on both sides of the BNLD were not significantly different between the PANDO and non-PANDO sides. No significant difference was observed in the incidence of paranasal abnormalities between sides. Only the minimum transverse diameter was significantly smaller on the PANDO side (3.63 ± 1.08 vs. 4.07 ± 1.13), and the septum deviated more toward the non-PANDO sides (67.9%).

Specific comments are below:

1. Only 3 figures and 3 tables are not enough for clearly investigating the topic described in this paper. Suggest add more CT data.

2. Maybe 3D reconstruction data may be closer to the truth and more convincing.

3. The control group for comparison is key to obtain the mentioned conclusion.

4. A total of 50 patients with unilateral PANDO (4 men and 46 women), whether such large differences in the number of men and women affected the results?

5. This paper also had lots of grammatical mistakes.

Round 2

Reviewer 1 Report (New Reviewer)

1. Abstract needs revision. Background starts with "To"....It is not complete. How many samples/images were tested? It has to be provided along with results (metrics) in the abstract.

2. Proof reading is essential.

3. Rest of the work is good.

Author Response

please see the attached file 

This manuscript is a resubmission of an earlier submission. The following is a list of the peer review reports and author responses from that submission.

Round 1

Reviewer 1 Report

Good paper

If the patients have been operated on, did the surgical reports corroborate the CT scan findings?

2 main surgical findings can happen: 1- mucosal inflammation causing narrow lumen 2- dilatation of the lachrymal ways. There is no reference to the sac or canal contents. Could the CT scan help in the diagnosis and anticipate pathologicak findings: lithiasis: fungal balls...

Could the CTscan help to differentiate wide meatus opening Vs submucosal path under the inferior meatus, knowing that asymmetries can be observed in a specific patient?

Reviewer 2 Report

The authors compared the morphology of the nasolacrimal canal between the affected and unaffected sides in unilateral PANDO. I think that this paper will not provide novel scientific findings because there have already been many similar papers previously. I have the following comments:

1. The authors diagnosed PANDO using lacrimal irrigation and probing. But a recent study showed less reliability of lacrimal irrigation for differentiation between nasolacrimal duct obstruction/stenosis and functional nasolacrimal duct delay (doi: 10.1007/s00417-022-05654-1.). Diagnosis using a dacryoendoscope, dacryoscintigraphy, and/or dacryocystography may be more reliable.

2. Why did the authors use paired t-test, although there was no dependent sample?

3. Generally, the word “duct” is used for mucosal passage, while the word “canal” is used for bony passage.

4. The authors discussed that a smaller BNLD diameter may be a risk factor for PANDO (line 203). However, they measured the bony canal diameter at a point of time, and the unaffected sides had a possibility of development of PANDO in the future. Therefore, they can not hypothesize that a smaller BNLD diameter is a risk factor for PANDO.

Round 2

Reviewer 2 Report

Thank you for revising the paper according to my comment. But as I mentioned previously, there have already been many aimilar papers previously. Therefore, I think that this paper will not provide novel scientific findings.